# Single-cell chromatin accessibility landscape in kidney identifies additional cell-of-origin in heterogenous papillary renal cell carcinoma

Qi Wang[1,8], Yang Zhang[1,2,8], Bolei Zhang[3], Yao Fu [4], Xiaozhi Zhao[5], Jing Zhang[1], Ke Zuo[2], Yuexian Xing[2], Song Jiang [2], Zhaohui Qin [6], Erguang Li[1], Hongqian Guo [5✉], Zhihong Liu [1,2✉] & Jingping Yang [1,2,7✉]

Papillary renal cell carcinoma (pRCC) is the most heterogenous renal cell carcinoma. Patient survival varies and no effective therapies for advanced pRCC exist. Histological and molecular characterization studies have highlighted the heterogeneity of pRCC tumours. Recent studies identified the proximal tubule (PT) cell as a cell-of-origin for pRCC. However, it remains elusive whether other pRCC subtypes have different cell-of-origin. Here, by obtaining genome-wide chromatin accessibility profiles of normal human kidney cells using single-cell transposase-accessible chromatin-sequencing and comparing the profiles with pRCC samples, we discover that besides PT cells, pRCC can also originate from kidney collecting duct principal cells. We show pRCCs with different cell-of-origin exhibit different molecular characteristics and clinical behaviors. Further, metabolic reprogramming appears to mediate the progression of pRCC to the advanced state. Here, our results suggest that determining cell-of-origin and monitoring origin-dependent metabolism could potentially be useful for early diagnosis and treatment of pRCC.

[1] Medical School of Nanjing University, Nanjing, Jiangsu 210093, China. [2] National Clinical Research Center for Kidney Disease, Jinling Hospital, Medical School of Nanjing University, Nanjing, Jiangsu 210002, China. [3] School of Computer Science, Nanjing University of Posts and Telecommunications, Nanjing, Jiangsu 210023, China. [4] Department of Pathology, Affiliated Drum Tower Hospital, Medical School of Nanjing University, Nanjing, Jiangsu 210008, China. [5] Department of Urology, Affiliated Drum Tower Hospital, Medical School of Nanjing University, Nanjing, Jiangsu 210008, China. [6] Department of Biostatistics and Bioinformatics, Rollins School of Public Health, Emory University, Atlanta, GA 30322, USA. [7] Jiangsu Key Laboratory of Molecular Medicine, Medical School, Nanjing University, Nanjing 210093, China. [8]These authors contributed equally: Qi Wang, Yang Zhang. ✉email: dr.ghq@nju.edu.cn; liuzhihong@nju.edu.cn; jpyang@nju.edu.cn

While pRCC patients with poor survival are known to display molecular features such as metabolic features, immunologic features, or CpG island methylator phenotype (CIMP)[1–3], it is unclear whether these features are the cause or outcome of tumor progression. From histological and molecular characterization studies, we know pRCC tumors are highly heterogenous and display various subtypes with different malignancies, making the disease extremely difficult to treat[1–4]. Studies of other cancers have shown that tumor heterogeneity may emerge from different cells-of-origin—normal cells afflicted with the cancer-causing mutation that go on to determine the fate and pathology of a tumor cell. In breast cancers[5] and glioblastoma[6,7], the same genetic mutation in a different cell lineage can lead to different morphology, phenotype and malignancy. Hematopoietic stem cells-derived leukemias showing higher methylation levels than granulocyte-macrophage progenitor cells-derived leukemias[8] also suggest that the molecular features of heterogenous tumors could be underpinned by cell-of-origin[5,7,9,10]. There is also evidence showing that cell-of-origin can affect tumor response to treatments[8,11,12]. Cell-of-origin, whose features are retained in evolving tumor cells[13], is therefore a powerful biomarker for early diagnosis. Screening cell-of-origin signatures in pRCC could potentially uncover unknown cells-of-origin that are linked to the various pRCC subtypes and their clinical behaviors. Understanding how the subtypes progress could help us develop more targeted treatments and reduce deaths[14].

Stable and heritable epigenetic signatures such as transcription factors programs[15], histone modifications[16] and DNA methylation[17] have been used effectively to trace the cell-of-origin and subtypes of morphological and molecular evolving tumor cells. In our study, we used another epigenetic signature, chromatin accessibility, to trace the cell-of-origin signatures in pRCC. Because chromatin accessibility reflects transcription factor binding, histone modifications and DNA methylation, it offers greater insights into the gene regulatory mechanisms of the cells. Additionally, the chromatin accessibility profiles of 34 human pRCC samples from The Cancer Genome Atlas (TCGA) have revealed the heterogeneity of pRCC[18].

In this work, we use single-cell assay for transposase-accessible chromatin sequencing (scATAC-seq) to capture the cell type-specific chromatin accessibility landscape of normal human kidney at single-cell resolution. The alignment of the ATAC-seq profiles of human pRCC samples against the landscape resolves heterogenous cell-of-origin of pRCC. Besides proximal tubule (PT) cells, pRCC can also originate from collecting duct principal cells (CD_PC) in the distal tubules of the kidney. We find that pRCCs with different cell-of-origin have different molecular characteristics, undergo different carcinogenic transformations, and display different clinical behaviors. pRCC also show cell-of-origin-dependent risks of progression, which is characterized by metabolism reprogramming. pRCC that originated from CD_PC are enriched for advanced pRCC and activated interferon signaling, suggesting their potential response to checkpoint immunotherapy. Our study demonstrates that the chromatin accessibility landscape is a potential early diagnostic method for advanced pRCC, allowing cell-of-origin dependent molecular characteristics to direct the treatment of pRCC.

## Results

### scATAC captures the epigenetic landscape of normal kidney.
We performed scATAC-seq on paracancerous kidney tissues from two patients undergoing nephrectomy to profile chromatin accessibility for cells in the kidney (Supplementary Fig. 1a, b). A total of 9460 high-quality nuclei were kept after stringent quality controls (Supplementary Fig. 1c, d), with a median read depth of 14,158 fragments per nuclei and median transcription start site (TSS) ratio (the percentage of fragments at TSS) of 0.15. After batch correction and unbiased clustering with Latent Semantic Analysis on term frequency-inverse document frequency (LSA-log (TF-IDF)), the nuclei formed 20 clusters (Supplementary Fig. 1e) with an even distribution in the samples (Supplementary Fig. 1f). Clusters with more than 50 cells were annotated by integrating the scATAC-seq with four human kidney single-cell RNA sequencing (scRNA-seq) datasets[19–22] (Supplementary Fig. 2a). Clusters that could not be consistently annotated (e.g., cluster 9) were re-clustered and re-annotated (Supplementary Fig. 2b, c). After integrating the scRNA-seq and examining the marker genes, a total of 20 clusters were finally annotated and all the main cell types in the kidney were identified (Fig. 1a and Supplementary Data 1). The annotation revealed a PT-related renal progenitor-like (PTPL) subgroup with chromatin accessibility at VCAM1 and PROM1 loci (Supplementary Fig. 2d). Further, the profiles display cell type-specific chromatin accessibility (Fig. 1b), notably at loci AQP1 in PT cells, which were not detected in scRNA-seq (Fig. 1c, d). Immunofluorescence staining of the kidney samples confirmed that high levels of AQP1 are expressed only in PT cells (Fig. 1e). These results demonstrate that scATAC-seq captured genome-wide transcriptional regulation with high sensitivity.

To investigate the cell type-specificity of epigenetic regulation, we characterized the chromatin accessible regions by comparing the scATAC-seq peaks of different cell types (Supplementary Fig. 3a). We detected 19,409 cell type-specific peaks and each cell type showed cell type-specific accessible regions (Fig. 1f and Supplementary Fig. 3b). A vast majority of these peaks were in non-coding regions (Supplementary Fig. 3c, d). Functional analysis of the specific peaks by GREAT[23] revealed that each group of cell type-specific peaks corresponds to the functions of the cell type (Fig. 1g). For instance, PT-specific peaks reflect their role in catabolism of small molecules while CD_PC-specific peaks, though small in number, reflect their specific role in urinary excretion. The cell type-specific peaks also revealed key transcription factors in each cell type (Fig. 1h). HNF4A, as a key transcription factor in PT cells[24], is highly accessible only in PT cells (Fig. 1i), and the genomic sites having the HNF4A motif only show activity in PT cells as well (Fig. 1j). These results indicate that the scATAC-seq profile sensitively captured cell type-specific epigenetic features in the kidney.

### Cell-of-origin of pRCC is heterogenous.
We used the established cell type-specific epigenetic features in kidney cells and the ATAC-seq of 34 human pRCC samples from TCGA identified by Corces et al.[18] to investigate the cell-of-origin of pRCC. To ascertain if cell type-specific epigenetic features can be used to reveal the cell-of-origin of cancers, we compared the cell type-specific peaks from our normal kidney scATAC-seq profile with the cancer type-specific peaks from pan-cancer chromatin accessibility profiles for 410 tumor samples across 23 cancer types identified in Corces et al.[18]. Renal cell carcinoma including pRCC showed significant overlap with kidney PT cells (Fig. 2a and Supplementary Fig. 4). This enrichment, consistent with previous reports that have identified pRCC tumor cells as originating from PT cells[19,25], confirmed that cell type-specific epigenetic features can be used to identify the cell-of-origin of cancers.

When the cell-of-origin for the 34 pRCC samples with ATAC-seq from TCGA[18] were subsequently examined individually, two different chromatin accessibility patterns were seen (Fig. 2b). While most of the pRCC samples (30 out of 34) show positive correlation with chromatin accessibility in the PT groups (with

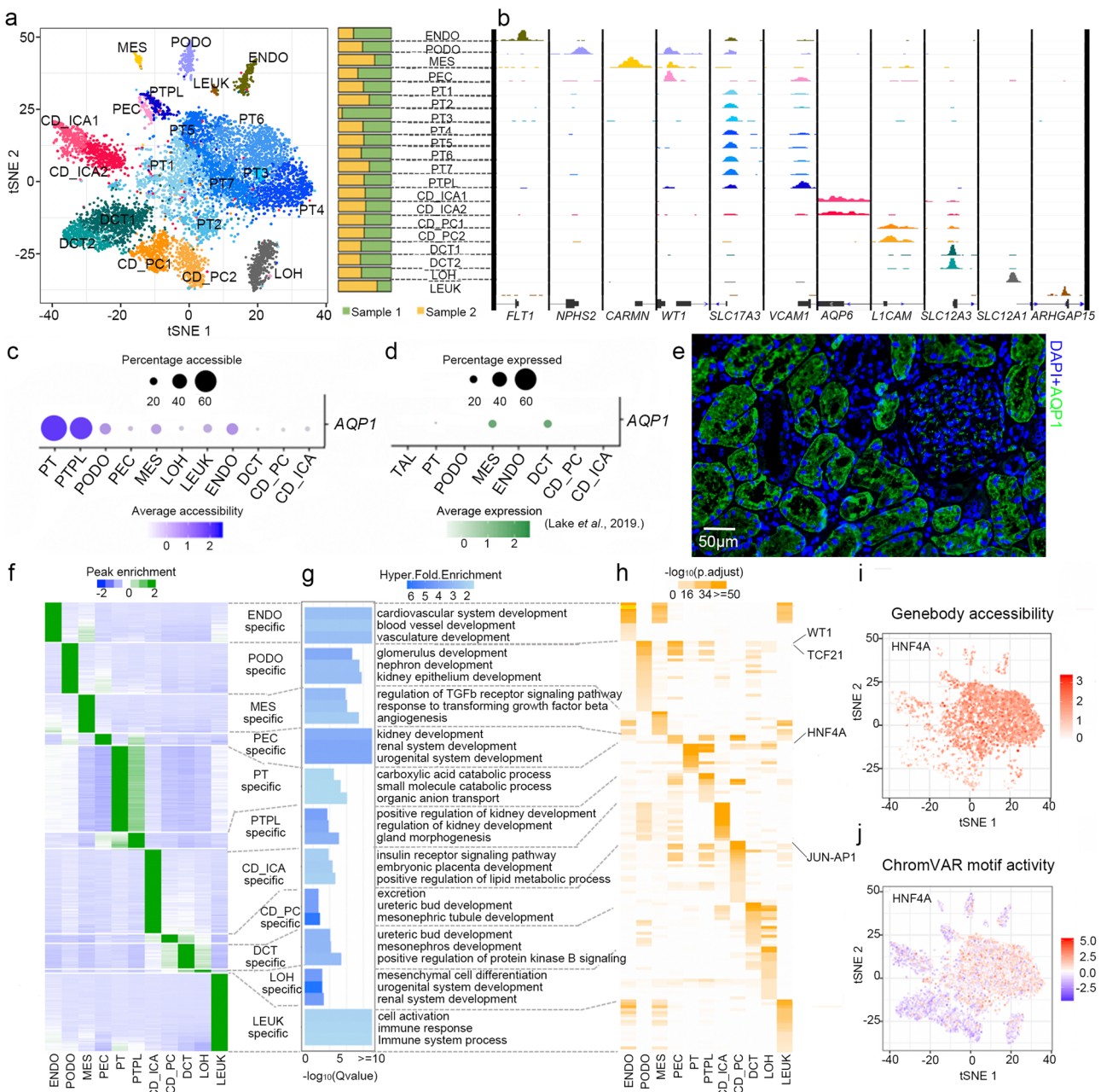

**Fig. 1 Single cell chromatin accessibility captures cell type-specific epigenetic features in human kidney. a** t-SNE embedding of 9428 kidney cells colored by clustering (left), and the proportions of each cluster in two samples (right). A total of 20 clusters with more than 50 cells were annotated. **b** Aggregated chromatin accessibility view shows cell type-specific chromatin accessibility at kidney cell type markers. **c** Genebody accessibility of *AQP1* in scATAC-seq. **d** RNA level of *AQP1* in scRNA-seq (Lake et al.[21]). **e** Immunofluorescence staining of paracancerous kidney tissues confirmed PT-specific expression of AQP1 (green) in kidney. Nucleus was stained with DAPI (blue). Data is from one experiment representative of three independent samples examined. **f** K-means clustering of 19,409 cell type-specific peaks with each cell type showing specific accessible regions. Each row represents a cell type-specific peak, and each column represents an individual cell type. **g** Enriched GO terms for cell type-specific peaks. Top three biological processes in each cell type are shown. **h** Heatmap of motif enrichment on cell type-specific peaks for each cell type. Each row represents a transcription factor motif and each column represents an individual cell type. Only motifs with -log10(p.adjust) > 10 in at least one cell type are shown. t-SNE of genebody accessibility (**i**) and chromVAR motif activity (**j**) of *HNF4A* in scATAC-seq show PT-specific expression and function of HNF4A. Source data are provided as a Source Data file.

highest correlation for PTPL), a small group (4 out of 34) show negative correlation with chromatin accessibility in PT (Fig. 2b). With a second quantitative method as similarity score[19], we found the exact same four samples showed strong similarity to distal tubule cell type CD_PC (Fig. 2c). In order to resolve the uncertainty from scATAC-seq clustering method, we also carried out clustering analysis using an alternative method cisTopic[26] (Supplementary Fig. 5), and we found all the findings obtained

using the two different clustering algorithms are highly consistent (Supplementary Fig. 6). We termed the first group as pRCC with accessibility pattern 1 (pRCCa1) and the second group as pRCC with accessibility pattern 2 (pRCCa2).

In an inverse strategy, where we identified pRCCa1- and pRCCa2-specific peaks before examining their accessibility in kidney cell types (Supplementary Fig. 7), the pRCCa2-specific peaks are indeed more accessible in distal tubule cell types,

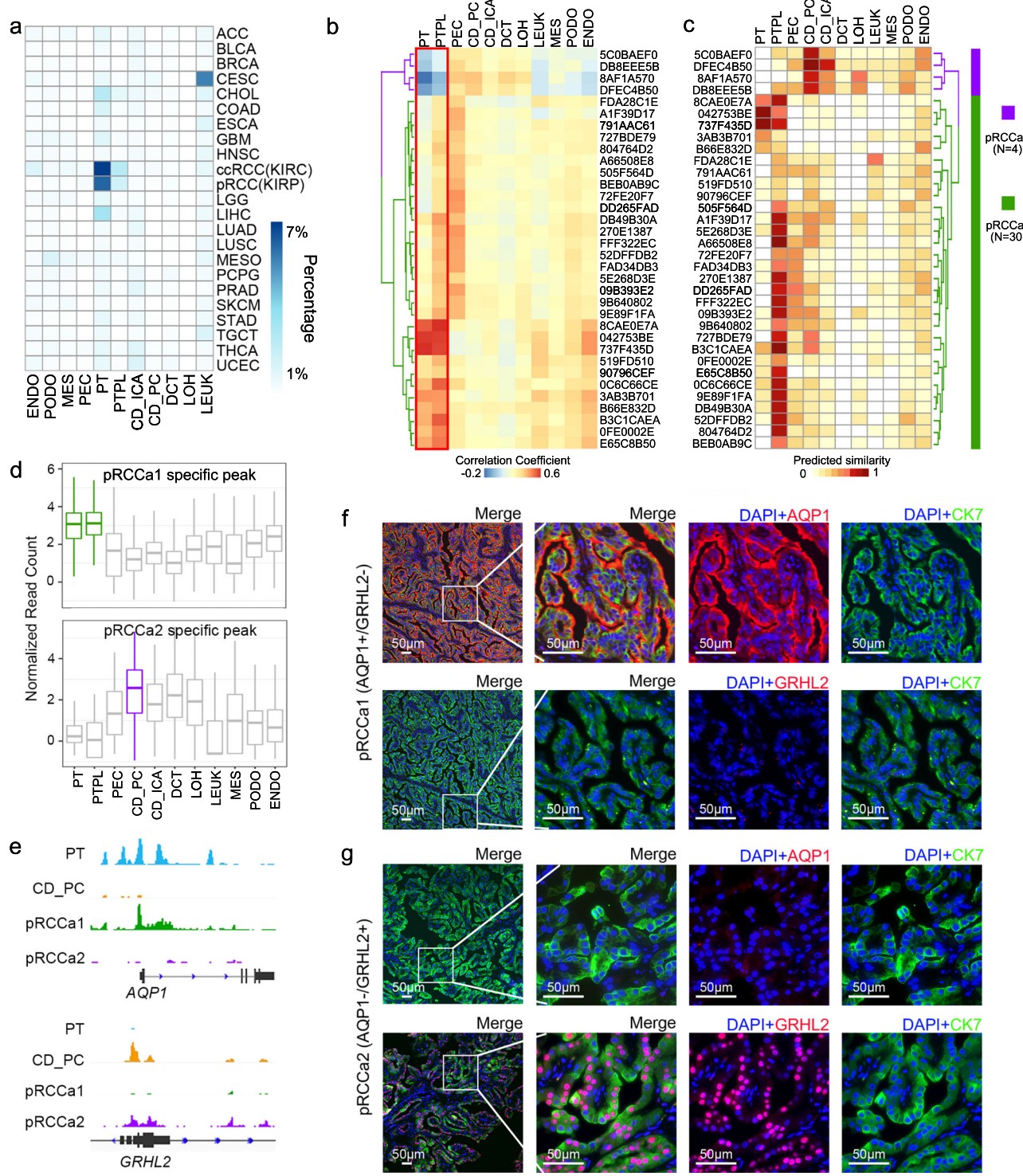

especially CD_PC (Fig. 2d). For example, pRCCa2 do not show chromatin accessibility at the PT marker *AQP1* locus, but are accessible at the CD_PC marker *GATA3* locus and at *GRHL2* locus which encodes a protein required for the function of collecting duct[27] (Fig. 2e and Supplementary Fig. 8a). Additionally, the pRCCa1/2-specific peaks are enriched with motifs for their respective cell-of-origin's key transcription factors (Supplementary Fig. 7). To verify this previously unknown cell-of-origin for pRCC, we recruited an independent pRCC cohort of 50 patients from Drum Tower hospital (Supplementary Fig. 1b) and

performed immunofluorescence staining with the PT marker AQP1, the CD_PC marker GATA3 or GRHL2, on tumor samples from these patients. Although most patients were AQP1+-GATA3− or AQP1+ GRHL2−, which indicates the tumors originated from PT cells (Fig. 2f and Supplementary Fig. 8b), 2 out of the 50 patients were AQP1-GATA3+ or AQP1-GRHL2+, suggesting a CD_PC origin (Fig. 2g and Supplementary Fig. 8c). Tumor cells with GATA3 or GRHL2 positive nuclei also contain the pRCC marker cytokeratin 7 (CK7) in the cytoplasm, without any traces of AQP1 (Fig. 2g and Supplementary Fig. 8c).

**Fig. 2 Integration of kidney scATAC-seq and 34 pRCC ATAC-seq from TGCA reveal the cell-of-origin of pRCC is heterogeneous. a** Heatmap showing the overlap between kidney cell type-specific peaks called from our kidney scATAC-seq and cancer type-specific peaks called from the pan-cancer ATAC-seq. Colors indicate the percentage of cancer peaks that overlap with kidney cell type-specific peaks. Pearson correlations (**b**) or similarity scores (**c**) between normalized chromatin accessibilities in 34 pRCC samples and normal kidney cell types identified with SnapATAC. Hierarchical clustering of pRCC samples based on the correlations or similarity scores revealed two subgroups (left), and heatmap view of the correlations or similarity scores show two different accessibility correlation patterns (pRCCa1 and pRCCa2) in the 34 pRCC samples. Each row represents a pRCC sample and each column represents a cell type. The red box highlights the PT cell type groups. The patients' prefixes of Stanford UUIDs are indicted for each sample. **d** Normalized read counts in kidney cell types at pRCCa1- (top) or pRCCa2 differential peaks (bottom) confirm pRCCa2 peaks are accessible in CD_PC. Box plots depict the median, quartiles and range. $n = 973$ peaks for pRCCa1 specific peak and $n = 175$ peaks for pRCCa2 specific peak. **e** Genome browser view in PT, CD_PC, pRCCa1, and pRCCa2 at PT marker, *AQP1* (top) or at CD_PC marker, *GRHL2* (bottom) loci. **f–g** Immunofluorescence staining of tumor tissues with AQP1 (red), GRHL2 (red), pRCC marker cytokeratin 7 (CK7, green) and DAPI (blue). Colocalization of AQP1 and CK7 and absence of GRHL2 confirmed PT-originated pRCC (**f**). Data are from one experiment representative of 48 independent experiments samples examined. Localization of GRHL2 in the cell nuclei with CK7 and absence of AQP1 confirmed CD_PC-originated pRCC (**g**). Data are from one experiment representative of two independent samples examined. Source data are provided as a Source Data file.

Together, these results demonstrate that the cell-of-origin of pRCC is heterogeneous and, in addition to the classical PT origin, we now identify distal tubule cells as an alternative cell-of-origin for pRCC.

**Cell-of-origin determines pRCC subtypes and their character.** Within TCGA are 255 additional pRCC samples without ATAC-seq. To examine the cell-of-origin for these pRCC samples, we transformed the different chromatin accessibility patterns into cell-of-origin feature genes (Fig. 3a). To do this, we optimized Cicero[28] to construct a regulatory network for each kidney cell type. We built the network based on genebody because its accessibility correlated better with transcriptional regulation than TSS accessibility[29] (Supplementary Fig. 9a). The optimized networks significantly improve regulatory prediction (Supplementary Fig. 9b) and show cell type-specificity (Supplementary Fig. 9c–f). We used our established regulatory networks for kidney cell types or a previously defined pRCC regulatory network[18] to extract the regulatory feature genes (Fig. 3a). Integrating pRCCa1 or pRCCa2 feature genes with their cell-of-origin feature genes, we obtained pRCC origin-derived features (Fig. 3a and Supplementary Fig. 10a). The features show both pRCCa1- or pRCCa2-specific and cell type-specific gene activity (Fig. 3b) as well as specific expression (Supplementary Fig. 10b, c).

Clustering the 255 TCGA pRCC samples based on expression of origin-derived features also revealed two groups, pRCCa1_Like (214 samples) and pRCCa2_Like (41 samples) (Fig. 3c). Most of those in the pRCCa2_Like group were pathological Type 2 pRCC (left panel in Fig. 3d) and 8 out of 41 displayed CIMP features that are present in advanced pRCC patients with worst survival (right panel in Fig. 3d). Interestingly, all CIMP samples in the pRCC cohort were unexpectedly characterized as pRCCa2 or pRCCa2_Like ($p = 3.9e\text{-}11$) (Fig. 3d and Supplementary Fig. 11). This suggests that the pRCCa2 cell-of-origin might be a predictor for CIMP. Trajectory analysis of the normal and pRCC samples showed both pRCCa1_Like and pRCCa2_Like diverged from the normal state and formed their own distinct paths, suggesting that they underwent different transformations (Fig. 3e). Comparing pRCCa1 or pRCCa2 with their corresponding cell-of-origin normal cells, we found that pRCCa1 was enriched with mTOR and NOTCH signaling pathways (Fig. 3f), consistent with a recent report that showed NOTCH overexpression in renal progenitor cells can lead to pRCC[25]. This finding suggests that inhibitors for NOTCH pathway[30] such as LY3039478 could potentially be used to reduce the incidence of pRCCa1. In contrast, pRCCa2 was mostly enriched with interferon-gamma and immune-related signaling pathways as shown by the high expression of *IL6* (Fig. 3g). Additionally, the two subgroups show distinct mutation patterns (Supplementary Fig. 12 and

Supplementary Data 2). Mutations in MET, PBRM1 and SMARCB1 only exist in pRCCa1_Like, while mutated genes enriched in pRCCa2_Like are enriched in NFR2 and HIPPO pathways. Together, these results indicate that cell-of-origin can segregate the pRCC cohort, and that pRCC with different cell-of-origin carry different molecular features and possibly underwent distinct transformation pathways.

**Clinical performance of pRCC depends on cell-of-origin.** To determine whether cell-of-origin is related to clinical performance of pRCC, we performed a disease association analysis using DisGeNET[31] and found that pRCCa2 origin-derived features are specifically related to tumor progression (Fig. 4a). Examining the clinical profiles confirmed that pRCCa2 or pRCCa2_Like displayed a higher distribution of late-stage tumor, higher metastasis to regional lymph nodes and worse survival than pRCCa1 or pRCCa1_Like (Fig. 4b, c and Supplementary Fig. 13a, b). When the effect of each origin-derived feature on clinical performance was examined, we found the panel of pRCCa2 origin-derived features predicted unfavorable prognosis, while pRCCa1 origin-derived features predicted favorable prognosis (Fig. 4d and Supplementary Fig. 13c–f). The prediction powers of pRCCa2 origin-derived features are even higher than other existing indicators (Supplementary Fig. 13g). Notably, these predictions are renal cell carcinoma-specific, especially for pRCC (Fig. 4e). In an alternative strategy, we extracted the top gene features for CIMP, the more lethal variants of pRCC, with logistic regression. We found these CIMP-associated gene features were highly expressed in CD_PC but not PT (Supplementary Fig. 14). Thus, the cell-of-origin of pRCC is related to their clinical performance.

The predicted poorer prognosis of pRCCa2_Like was in line with the finding that all CIMP were characterized as pRCCa2_Like (Fig. 3d). When we inspected the trajectory of pRCC, we found pRCCa2_Like_CIMP tended to locate at the end of the pRCCa2_Like branch while pRCCa2_Like_NotCIMP were found between CIMP and normal samples (Fig. 4f). Survival analysis revealed that pRCCa2_Like_NotCIMP performed better than pRCCa2_Like_CIMP but remained worse than pRCCa1_Like (Supplementary Fig. 15a). The progression from the normal state to the terminal of this branch was accompanied by a gradual increase in death and tumor metastasis to regional lymph nodes (Supplementary Fig. 15b). These results suggest that CIMP is likely an advanced state in the pRCCa2_Like subgroup.

To understand the factors driving pRCC progression in samples with the same cell-of-origin, we compared pRCCa2_Like_NotCIMP with pRCCa2_Like_CIMP and found that metabolic pathways were disrupted. The oxidative phosphorylation pathway was mostly disrupted between the two states (Supplementary Fig. 16a). The most up-regulated gene was lactate dehydrogenase A

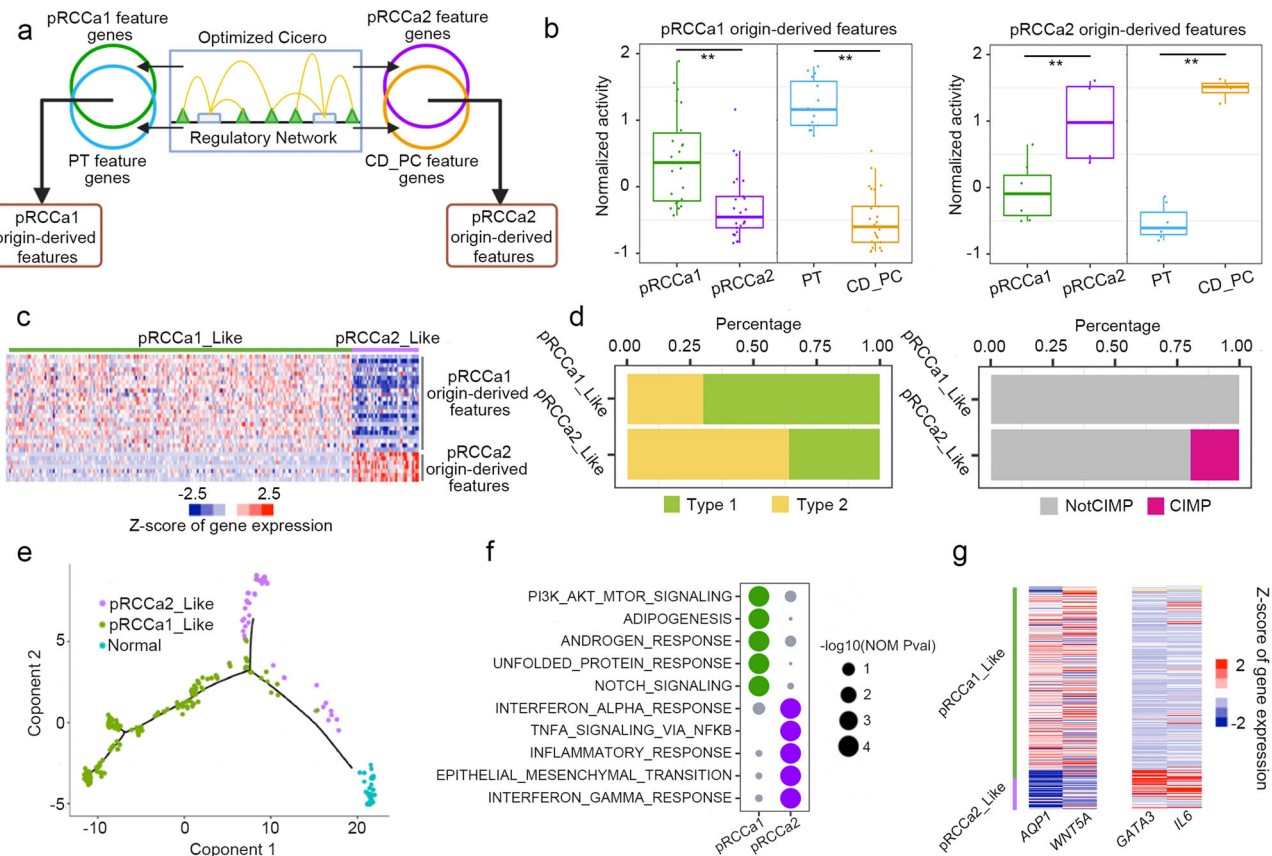

**Fig. 3 pRCC with different cell-of-origin show different characteristics. a** Strategy used to obtain origin-derived feature genes from chromatin accessibility. Optimized Cicero was used to construct a regulatory network from chromatin accessibility profiles. Regulatory networks are used to extract feature genes. Overlapping feature genes in pRCC subgroups and their respective cell-of-origin produce the origin-derived features. **b** Normalized gene activities of origin-derived features in pRCCa1, pRCCa2, PT and CD_PC. Gene activities were calculated based on the regulatory networks. Box plots show distribution of gene activities of origin-derived features. Each dot represents an individual feature gene. Feature genes show the pRCC subgroup and kidney cell type-specific activities. Box plots depict the median, quartiles and range. $n = 25$ genes for pRCCa1 origin-derived features and $n = 7$ genes for pRCCa2 origin-derived features. $p = 3.97e-09$ between pRCCa1 and pRCCa2 and $p = 6.24e-16$ between PT and CD_PC on differential activity of pRCCa1 origin-derived features. $p = 0.009$ between pRCCa1 and pRCCa2 and $p = 0.0007$ between PT and CD_PC on differential activity of pRCCa2 origin-derived features. Two-sided paired $t$ tests were used. ** indicates $p < 0.01$. **c** Heatmap of gene expression of origin-derived features for 255 TCGA pRCC samples show two expression patterns, pRCCa1_Like and pRCCa2_Like. Colors represent Z-scores of RNA-seq gene expression across the samples. Each row represents a feature gene and each column represents a pRCC sample. **d** Percentage of pathological Type 1/Type 2 (left) or CIMP (right) in pRCCa1_Like or pRCCa2_Like samples. **e** Trajectory of the 255 TGCA pRCC samples and 32 normal kidney samples show pRCC diverged into distinct branches. Samples are colored according to their identities. **f** Top 5 Gene Set Enrichment Analysis (GSEA) hallmark terms enriched in pRCCa1 or pRCCa2 compared with their respective cell-of-origin based on gene activity. **g** Expression of representative genes in 255 TCGA pRCC samples in the same order as **c**. AQP1: feature gene for pRCCa1 with PT origins; WNT5A: representative gene in Notch pathway; GATA3: feature gene for pRCCa2 with CD_PC origins; IL6: representative gene in Interferon-gamma pathway. Colors represent Z-scores of RNA-seq gene expression across the samples. Source data are provided as a Source Data file.

(*LDHA*), which encodes an enzyme relevant to metabolism that converts glucose into energy, while the most down-regulated gene was fumarate hydratase (*FH*), which encodes an enzyme in the Krebs cycle that allows cells to use oxygen to generate energy (Fig. 4g). Survival analysis further showed that high expression of *LDHA* predicted the unfavorable prognosis for pRCCa2_Like (Fig. 4h). However, the association between *LDHA* and poor prognosis only existed in pRCCa2_Like, but not in pRCCa1_Like (Supplementary Fig. 16b). We then tried to develop a model to predict CIMP. With origin-derived features and LDHA, we trained a Random Forest model on the 34 pRCC samples where we identified the origin-derived features, and tested it on the remaining 255 pRCC samples. The model performed well on the identification of CIMP with AUROC at 0.98 (Supplementary Fig. 17). These results demonstrate that cell-of-origin dictates the prognosis of the clinical behaviors of pRCC. In the case of pRCCa2,

metabolic reprogramming appears to mediate its progression to the advanced state as CIMP and should therefore be closely monitored as a preventive measure (Fig. 4i).

## Discussion

Previous study indicated that some of the pRCC samples are not similar to the proximal nephron with transcriptional reference from micro-dissected nephrons[3], but the limitation in the data resolution made resolving their cell-of-origin at cell type level almost impossible. Using scATAC-seq technology, we constructed the epigenetic landscape of normal kidney cell types at single-cell resolution and used the landscape to resolve the cell-of-origin of pRCC. We discovered that the cell-of-origin of pRCC is heterogenous in that, besides the classical PT cells, we identified CD_PC in the distal tubule of the kidney as an alternative origin.

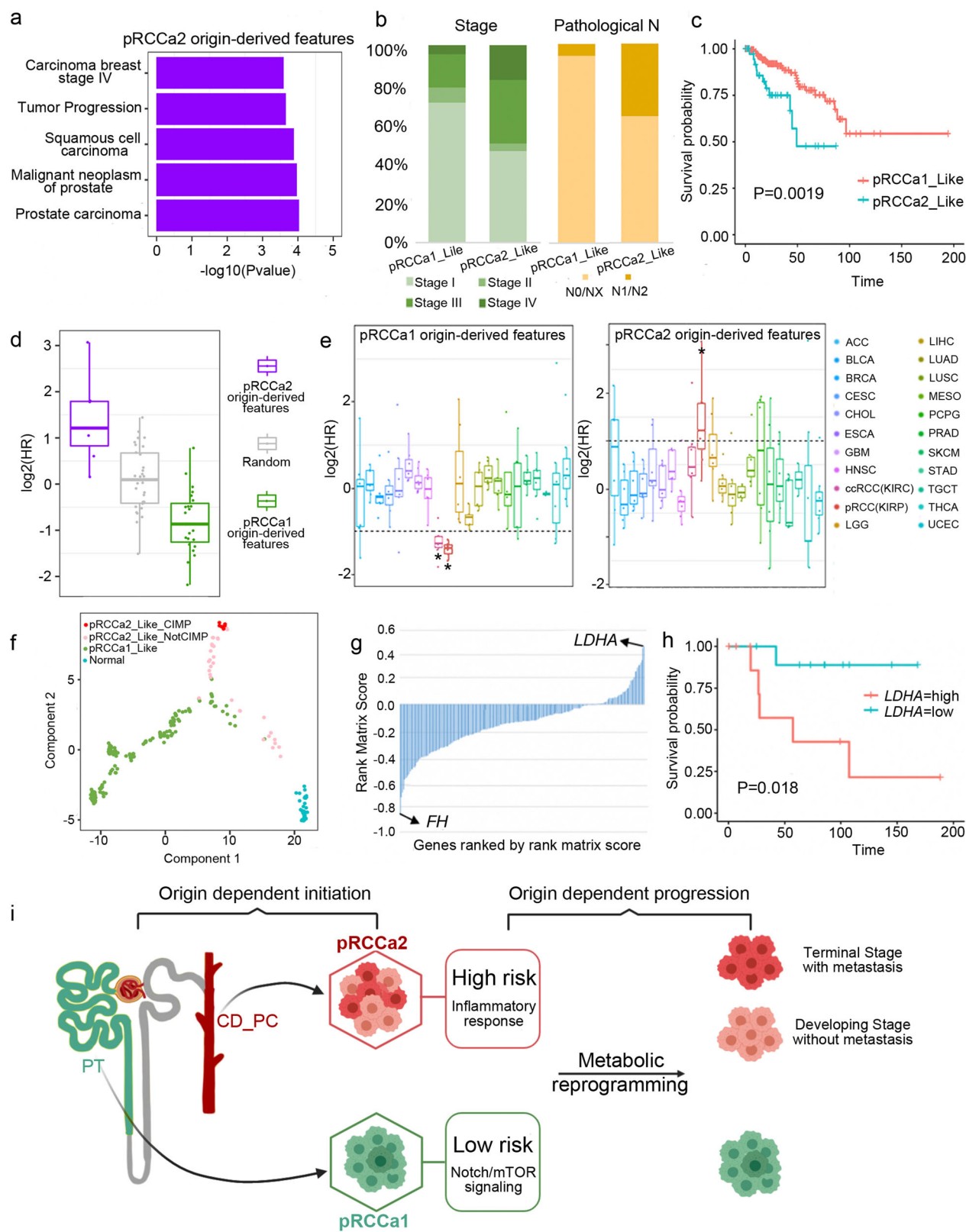

Unlike pRCC with PT origins where NOTCH and mTOR signaling pathways dominate, pRCC originating from CD_PC was enriched for inflammation and immune-related pathways. Our results further show that pRCC with CD_PC origins have a higher risk of progressing to advanced pRCC than those with PT origins, and this progression appears to be driven by metabolic reprogramming. Previous studies have identified metabolic and immunologic features for CIMP, the more lethal pRCCs variants[2,3]. However, these features are dynamic during tumor progression. For example, ccRCC aggressiveness is linked to increases in metabolites involved in glutathione and cysteine/methionine metabolism pathways[32]. Although these features can inform the more lethal state of pRCC, but they are unable to serve as prognostic biomarkers for early detection of the potential

**Fig. 4 Cell-of-origin predicts clinical behavior of pRCC. a** Top 5 enriched DisGeNET terms for pRCCa2 origin-derived features suggest that pRCC with these features have poor clinical performance. **b** Distribution of tumor stage (green), and percentage of metastasis to regional lymph nodes (orange) in pRCCa1_Like or pRCCa2_Like samples. N: regional lymph nodes. N0: No regional lymph node metastasis; NX: Regional lymph nodes cannot be assessed; N1/N2: Metastasis in regional lymph node(s). **c** Overall survival for pRCCa2_Like samples ($n = 41$) is poorer than pRCCa1_Like ($n = 214$). The Log-rank test was used to assess the univariate survival differences. **d** Hazard ratio from survival analysis for pRCCa2 and pRCCa1 origin-derived features and random selected genes. Boxplot shows distributions of hazard ratios for each group of feature genes. pRCCa2 origin features predicted unfavorable prognosis while pRCCa1 origin features predicted favorable prognosis. Each dot represents an individual gene. Box plots depict the median, quartiles and range. $n = 25$ genes for pRCCa1 origin-derived features, $n = 7$ genes for pRCCa2 origin-derived features and $n = 30$ genes for random selected features. **e** Hazard ratio from survival analysis for pRCCa1 (left) or pRCCa2 (right) origin feature panel. Survival analysis is performed for individual cancer types. Dash lines indicate $\log2(HR) = -1$ or $\log2(HR) = 1$. Box plots depict the median, quartiles and range. $n = 25$ genes for pRCCa1 origin-derived features and $n = 7$ genes for pRCCa2 origin-derived features. * denotes the cancer type with median $\log2(HR) > 1$ or $< -1$. **f** Same trajectory as Fig. 3e but with pRCCa2_Like_CIMP and pRCCa2_Like_NotCIMP identified. **g** Gene expression in the oxidative phosphorylation pathway show LDHA was upregulated while FH was down-regulated. Data is ranked by GSEA rank matrix score (pRCCa2_Like_CIMP versus pRCCa2_Like_NotCIMP). **h** Kaplan-Meier analysis of overall survival of pRCCa2_Like stratified by *LDHA* expression. *LDHA* gene expression was divided into "high" and "low" representing the upper and lower quarter of *LDHA* gene expression, respectively. The Log-rank test was used to assess the univariate survival differences. **i** Schematic shows pRCC initiated from different cell-of-origin possesses different molecular characteristics and presents different clinical performance. pRCCa2 with CD_PC origins display a higher risk of progression to terminal stage than pRCCa1 with PT origins. Metabolic reprogramming leads to the progression in pRCCa2. Source data are provided as a Source Data file.

aggressive pRCC. In contrast, cell-of-origin is stable and retained in the evolving tumor cells, and is therefore a much more reliable prognostic biomarker for early diagnosis.

Our study has several implications. Using ATAC-seq technology to identify the cell-of-origin of pRCC at the outset enables us to better diagnose the risk of progression and plan treatments. Further, monitoring metabolic changes could be an efficient way to track the progression of pRCC. Capturing the etiology of the disease early on can substantially improve treatment and patient survival[33]. For advanced pRCC where no standard-of-care systemic therapy exists[34], identifying the cell-of-origin of pRCC provides insights on the extent of the heterogeneity of the disease, which could be used to develop better targeted treatments. For instance, insights from our study suggest that metabolism-directed therapy could potentially be effective for preventing the progression of pRCC to the advanced stage while NOTCH inhibitors are better for pRCC with PT origins.

Importantly, our results also explain why current treatments for pRCC such as VEGF- and mTOR-directed therapies[35] and addition of interferon-α[36] show only marginal clinical benefits. As shown in our study, most advanced pRCC originate from CD_PC that display signatures related to immune response. Instead of NOTCH and mTOR signaling that are related to aberrant cell proliferation, advanced pRCC shows a pre-existing adaptive immunity. This suggests that using immune checkpoint inhibitors are likely to offer better clinical outcomes for advanced pRCC. Indeed, recent clinical trials demonstrated that the PD-L1 inhibitor Atezolizumab improved the objective response rate in a group of advanced renal carcinoma with immune signatures[37]. Thus, identifying cell-of-origin can predict differential clinical outcomes that will require different therapies for the different stages of pRCC.

## Methods

**Tissue procurement**. This study was approved by Jinling Hospital and Drum Tower Hospital Affiliated with the Medical School of Nanjing University under an established Institutional Review Board protocol. We have complied with all relevant ethical regulations with human participants. Informed consent for the human study was obtained by all participants. Paracancerous kidney tissues and tumor tissues were obtained from patients diagnosed as pRCC and undergoing nephrectomy for renal tumor.

**Nuclei isolation and scATAC-seq library preparation**. Kidney tissues were cut into 1 mm³ segments and underwent cell lysis using a Dounce homogenizer (885302–0002; Kimble Chase) in 1 ml chilled Lysis Buffer (10 mM Tris-HCl (pH 7.4), 10 mM NaCl, 3 mM MgCl₂, 0.1% Tween-20, 0.1% Nonidet P40 Substitute, 0.01% digitonin and 1% BSA) on ice. 9 ml chilled Wash Buffer (10 mM Tris-HCl

(pH 7.4), 10 mM NaCl, 3 mM MgCl2, 0.1% Tween-20 and 1% BSA) was then added and the resulting solution was filtered through a 30-μm cell strainer (130-098-458; Miltenyi Biotec). Nuclei were collected by centrifuging (500 g for 5 min at 4 °C) and resuspended in chilled Diluted Nuclei Buffer (10x Genomics; 2000207) at ~3000–4000 nuclei per μl. The nuclei concentration was determined using a Countess II FL Automated Cell Counter.

scATAC-seq libraries were prepared according to manufacturer protocol of Chromium Single Cell ATAC Library Kit (10x Genomics, PN-1000087). Briefly, the required number of nuclei were combined with ATAC Buffer B and ATAC Enzyme to form a Transposition Mix which was incubated at 37 °C for 60 min. Then the premix containing Barcoding Reagent B, Reducing Agent B and Barcoding Enzyme, the transposed nuclei, the Single Cell ATAC Gel Beads and partitioning Oil was loaded onto a Chromium Chip E. Resulting single-cell GEMs were collected and linear amplification was conducted in a C1000 Touch Thermal cycler as 72 °C for 5 min, 98 °C for 30 s, cycled 12×: 98 °C for 10 s, 59 °C for 30 s and 72 °C for 1 min. Emulsions were then coalesced using the Recovery Agent and cleaned up using Dynabeads. Indexed sequencing libraries were then constructed, purified and sequenced on an Illumina NextSeq instrument at Annoroad Gene Technology Company (Beijing, China).

**Raw data processing and quality control of scATAC-seq**. Barcode sequences consisting of 16 bp were obtained from the I2 index reads. For each read pair, we appended the barcode sequence to the read name. We checked the barcode against the barcode whitelist, and only kept the read pairs whose barcode perfectly matched any barcodes in the whitelist. We next trimmed the read pairs passing barcode checking for adapter sequences using trimmomatic[38] (0.32) with options "TRAILING:3 SLIDINGWINDOW:4:10 MINLEN:25". The trimmed read pairs were aligned to hg38 reference genome using Bowtie2[39] (2.3.0) with options "-X 2000 -3 1", and aligned fragments with a mapping quality less than 30 or that did not map uniquely to autosomes or sex chromosomes were filtered out using Samtools[40] (1.3.1). We subsequently removed PCR duplicates (fragments with identical start and end positions) on a cell-by-cell basis.

We identified barcodes representing genuine cells based on three criteria: (1) number of total fragments count; (2) the ratio of fragments in TSS region; (3) periodicity in the frequency of insert sizes. TSS region was defined as $+/-1000$ bp from TSS (RefSeq of hg38). Periodicity was measured as Cusanovich et al.[41]. Briefly, a periodogram was calculated using a fast Fourier transform of insert sizes for each barcode using the "spec.pgram" in R (3.5.1) with parameters "pad = 0.3, tap = 0.5, span = 20" and the spectral densities for frequencies between 100 and 300 bp were summed as banding score for the measure of periodicity. The $\log10$(numbers of total fragments), the ratios of fragments in TSS region, and $\log10$(banding scores) showed bimodal distribution. We used both the mclust[42] package in R and view by plot to determine the cutoff of the criteria, and selected the more stringent one as thresholds. The barcodes retained as cells were those with $\log10$ (number of total fragments) $> 3.4$, the ratio of fragments in TSS region $> 0.15$ and $\log10$ (banding score) $> -1.75$.

**Clustering analysis for scATAC-seq**. The clustering analysis were conducted using LSA-log (TF-IDF) in SnapATAC[43] (1.0.0) or cisTopic[26]. For SnapATAC, we split the genome into 5 kb windows and removed windows overlapping ENCODE blacklisted regions (http://mitra.stanford.edu/kundaje/akundaje/release/blacklists). We counted fragments for each window in each cell, converted non-zero value to 1, and generated a large, sparse, binary window-by-cell matrix. Based on this matrix, we filtered out the top 5% of windows with abnormally high coverage and windows

with no coverage. We then used "Log TF-IDF" transformation to reduce the dimensionality of this large binary matrix, and used harmony[44] to remove batch effect. We used the 2nd through 40th dimensions to construct the k-nearest neighbor (KNN) Graph with the option "$k = 15$", which was subsequently used for Louvain[45] clustering with a default resolution of 1 and for Rtsne[46] visualization. Re-clustering was performed the same way as above. All clusters with more than 50 cells are retained for downstream analysis. For cisTopic, the filtered binary window-by-cell matrix created above was used to perform Latent Dirichlet Allocation, and the resulted topics-by-cells probability matrix was used for downstream analyses with default parameters.

**Identifying peaks for scATAC-seq**. To identify peaks for each cell cluster, we aggregated unique aligned fragments associated with cells in a given cluster. We then used MACS2[47] (2.1.1) to call peaks on the aggregated profiles with parameters "–nomodel –keep-dup all –extsize 200 –shift 100 –nolambda -q 0.001". Peaks overlapping ENCODE blacklisted regions were removed to generate clean peaks for each cluster.

A master peak list was generated by taking all clean peaks identified in any cluster and merging them with bedtools[48] (2.25.0).

High quality peaks were selected as peaks from master peak list with accessibility in more than 5% of the cells in any cluster.

To identify high variable peaks and cell type-specific peaks, we generated a peak-by-cell-type proportion matrix. We calculated the proportion of cells of each cell type that were accessible at each high quality peak. High variable peaks were identified based on the peak-by-cell-type proportion matrix using "FindVariableFeatures" in Seurat[49] with parameters "selection.method = "vst", nfeatures=50,000".

Cell type-specific peaks were identified using the R script developed by Cusanovich et al.[41]. Briefly, we truncated peak-by-cell-type proportion matrix for the 50,000 high variable peaks. We then multiplied the matrix for each cell type with a scaling factor, which was the median fragment number for cells in each cell type, to arrive at cell type normalized proportion matrix. The normalized proportion matrix was used to calculated Jensen-Shannon-based specificity scores. We then squared these scores and multiplied them again by the normalized proportion matrix to produce final specificity scores. We selected top 20,000 returns ranked by final specificity scores, which were represented by 19,409 cell type-specific peaks. K-means clustering of cell type-specific peaks were performed with Cluster3[50] and visualized with Treeview[51]. Genomic annotations for cell type-specific peaks were performed with ChIPseeker[52], and gene ontology enrichment analysis of cell type-specific peaks were performed with GREAT[23].

**Normalized bigwigs and sequencing tracks**. To visualize scATAC-seq cluster data genome-wide, we created ATAC-seq signal tracks that were normalized by the number of fragments in master peaks. We converted the bedGraph file for each cluster into bigwig file using "bedGraphToBigWig" tool. We then counted the number of fragments in master peaks for each cluster from the bam file using "bamfile.fetch" in Python (2.7.10) and used the number as scale factor for each cluster. The bigwig files were viewed in IGV. All track figures in this paper show groups of tracks with matched scale factors.

**Cell type annotation of scATAC-seq clusters**. To annotate cell type identity for each cluster, we integrated scATAC-seq with scRNA-seq using Seurat[53]. Four scRNA-seq datasets of human kidney were obtained from GSE140989[20], GSE121862[21], GSE131882[22], or downloaded from the supplementary files of Young et al. [19]. For the scRNA-seq datasets without author-provided annotation as GSE140989 and GSE131882, we re-analyzed the dataset by clustering using scRNA-seq pipeline in Seurat[49] (3.0.0) and annotating with markers reported in respective manuscript. With the scRNA-seq annotation results, we followed the working flow of 'Integrating scRNA-seq and scATAC-seq data' in Seurat to annotate scATAC-seq clusters. Briefly, we constructed a peak-by-cell count matrix for master peaks and collapsed the peak-by-cell count matrix using "CreateGeneActivityMatrix" to a genebody accessibility-by-cell matrix which was subsequently used for unsupervised identification of anchors between scATAC-seq and scRNA-seq with "FindTransferAnchors". These anchors were then used to transfer the cell type labels learned from scRNA-seq to scATAC-seq cells. We calculated the proportion of each transferred scRNA-seq label type in each scATAC-seq cluster. The scRNA-seq label type with highest proportion was assigned to the scATAC-seq cluster. The cell type assignments were accepted if they were consistent in at least 2 out of the 4 scRNA-seq datasets.

For clusters without successful assignment through above integration strategy, we then manually annotated them based on commonly used marker genes for kidney cell types[22,54–56].

**Motif analysis**. To identify key regulators of chromatin accessibility, we performed motif enrichment analysis using HOMER[57] (4.11.1) on cell type-specific peaks with parameter "-size given". In each cell type, the $-\log10$(p.adjust) for each known motif was calculated as $-\log10$(p values) multiplying 1000 and then corrected by dividing the sum of the calculated values (for p value = 0, the calculated value was

set to 200). Motifs with $-\log10$(p.adjust) > 10 in at least one cell type were used for clustering and visualization.

In addition to HOMER, we measured global TF activity using chromVAR[58] (1.4.1), based on peak-by-cell count matrix. We corrected GC bias based on "BSgenome.Hsapiens.UCSC.hg38" using "addGCBias". The filtered collection of human motifs from CIS-BP database was used and the deviation z-scores for each TF motif in each cell were calculated using "computeDeviations" with default parameters.

**Comparing normal kidney scATAC-seq with cancer ATAC-seq**

*Comparison by peak overlapping*. The comparison of chromatin accessibility between normal kidney cell type and pan-cancer types was performed based on overlapping of peaks. The peaks for each normal kidney cell type were the cell type-specific peaks defined above. The peaks for each cancer type were downloaded from https://gdc.cancer.gov/about-data/publications/ATACseq-AWG[18]. We overlapped peaks from each cancer type with peaks from each normal kidney cell type using "findOverlaps" in GenomicRanges. We then calculated the percentage of peaks from each cancer type that overlapped with peaks from each normal kidney cell type. The significance of overlapping was calculated with a permutation test. For each overlap between a cell type and a cancer type, we randomly select the same number of peaks for the cell type from the pooled cell type-specific peaks, count the number of the cancer type-specific peaks that overlap with the randomly select cell type peaks, and calculate the percentage. We repeat these steps 1000 times, and the average of the 1000 percentages is taken as the expected overlapping percentage. p value is calculated using binomial test with the "binom.test" function in R. All the p-values are then adjusted using the Benjamini–Hochberg procedure with the "p.adjust" function in R.

*Comparison by correlation*. The comparison of chromatin accessibility between normal kidney cell types and 34 pRCC samples was performed based on correlation of accessibility matrix. Normalized panpeak-by-sample matrix for 34 pRCC samples was downloaded from https://gdc.cancer.gov/about-data/publications/ATACseq-AWG[18]. As each sample had two replicates, we took the average of the two replicates to represent each sample and constructed a 562,709 × 34 matrix for pRCC. In order to be comparable with the 34 pRCC samples, we constructed a normalized panpeak-by-cell matrix for scATAC-seq the same way as described in Corces et al.[18]. We used the 562,709 peaks from pan-cancer ATAC-seq which covered 84% of scATAC-seq master peaks to generate the matrix. We adjusted the start and end of the fragments ("+" stranded +4 bp, "−" stranded −5 bp) using "scanbam" in Rsamtools, obtained the number of fragments per peak for each cluster using "countOverlaps", and compiled a 562,709 × 20 matrix. The matrix was then normalized using "cpm" in edgeR (3.24.3) with parameters "log = TRUE, prior.count = 5" followed by a quantile normalization using "normalize.quantiles" in "preprocessCore".

Based on these two normalized matrices, we next calculated the correlation of accessibility between normal kidney cell types and pRCC individual samples. Briefly, we used pRCC matrix to identify high variable peaks for pRCC ATAC-seq using "FindVariableFeatures" in Seurat with parameters "selection.method = "vst", nfeatures = 50000". The high variable peaks for normal kidney scATAC-seq were defined above. The overlapping high variable peaks were used to reduce the two normalized matrices to 15,390 × 34 and 15,390 × 20 dimensions. Pearson correlation coefficient was calculated for pairs of each normal kidney cell type and each pRCC sample using "cor" in R and the average correlation coefficient from cell subtypes was taken for each cell type group.

*Comparison by similarity score*. In order to predict the similarity scores for bulk pRCC ATAC-seq from the normal kidney scATAC-seq data, we generated pseudo-bulk reference panel from kidney scATAC-seq data. We randomly sample 50 cells 20 times from each cell type to generate the pseudo-bulk reference panel. The full 562,709 × 220 matrix of pseudo-bulk reference for the 11 cell type groups was converted from the panpeak-by-cell matrix by sum up the counts from the 50 cells. The pseudo-bulk reference matrix then normalized using "cpm" in edgeR (3.24.3) with parameters "log = TRUE, prior.count = 5" followed by a quantile normalization using "normalize.quantiles" in "preprocessCore". Cell type-specific peaks from normal kidney scATAC-seq and high-quality peaks from pRCC ATAC-seq (top 80,000 ranked) were used to reduce the pseudo-bulk reference matrix and panpeak-by-sample pRCC matrix to 15,081 × 220 and 15,081 × 34 dimensions, respectively.

We then trained binomial logistic regression models on the cellular identities in the pseudo-bulk reference, and used these models to predict similarity scores for each pRCC sample. This approach was implemented using the R package glmnet[59] (4.1-1). In the training process, we used an offset for each model as $\log(f/(1 - f))$ (f is the fraction of cells in the cell type being trained), performed tenfold cross validation and selected the largest regularization coefficient, lambda. In the predicting process, an offset of 0 was used.

*Differential peaks between pRCC subgroups*. The differential peaks between pRCC subgroups (pRCCa1 and pRCCa2) were identified by edgeR (3.24.3) as abs(logFC) > 2.5, FDR < 0.05, and logCPM > 1. The accessibilities of these peaks in

normal kidney cell types were then compiled from above normalized panpeak-by-cell matrix.

*Optimization of Cicero and computing gene activity*. We optimized Cicero[28] (1.0.15) by using "genebody accessibility" instead of "TSS accessibility" to calculated co-accessibility scores. Briefly, we computed "genebody accessibility" for each protein_coding gene (ensemble hg38) in each cell by summarizing the overall chromatin accessibility of all peaks within a given gene based on the binary peak-by-cell count matrix. The 'genebody accessibility' matrix were assigned with rownames as −200/+100 bp coordination of TSS, and was appended to the original binary peak-by-cell count matrix. We then used "ran_cicero" function in Cicero on this matrix for each cell type to establish co-accessible connections using parameters "aggregation $k = 30$, window size $= 500$ kb, distance constraint $= 250$ kb". Cutoff as 0.1 were used to establish the regulatory networks.

Gene activities were calculated using "build_gene_activity_matrix" in Cicero. For normal kidney cells, the above regulatory networks were used. For pRCC samples, the cancer regulatory networks (https://api.gdc.cancer.gov/data/dbf21e32-c335-4d86-b42b-f0d5d1808fef)[18] were used.

*Origin-derived features*. We identified feature genes for normal kidney cell types and pRCC subgroups based on gene activity matrices compiled above. Briefly, for normal kidney cells, we filtered out genes that were accessible in less than ten cells, and extracted cell type feature genes with "pct.1 > 0.5, avg_logFC > 0.5" using "FindAllMarkers" in Seurat. For pRCC subgroups, we extracted feature genes with "abs (logFC) > 1, FDR < 0.05, logCPM > 5" using edgeR. We then used overlapping of PT feature genes and pRCCa1 feature genes as pRCCa1 origin-derived features, and overlapping of CD_PC feature genes and pRCCa2 feature genes as pRCCa2 origin-derived features. DisGeNET[31] enrichment analysis of origin-derived features was performed using Enrichr[60].

*Clustering of pRCC samples*. Clustering of pRCC was performed using Cluster3, based on correlation with PT cell types for 34 TCGA pRCC samples with ATAC-seq, or based on RNA-seq gene expression of origin-derived features for 255 pRCC samples without ATAC-seq. RNA-seq gene expression (FPKM) of pRCC were downloaded from UCSC XENA[61]. Clinical information of pRCC was downloaded from Ricketts et al.[2].

*Trajectory analysis of 255 pRCC samples*. Trajectory analysis of 255 TCGA pRCC samples along with normal kidney samples was performed using Monocle 2[62] (2.10.1) pseudotime ordering algorithm. Briefly, we filter out genes with average expression < 1 or deviation of expression > 10 in the gene expression matrix consisting of 214 pRCCa1_Like samples, 8 pRCCa2_Like_CIMP samples, 33 pRCCa2_Like_NotCIMP samples and 32 normal kidney samples. The filtered matrix subsequently underwent t-SNE dimensionality reduction. Differential expression test was performed for each gene, including the sample labels as indicator variables to find differential expression genes across samples. The differential expression genes with p-value < 0.00003 were chosen as ordering genes for trajectory inference. We then used Monocle 2's DDRTree dimensionality reduction algorithm to align samples to a branched trajectory.

*Top gene features associated with CIMP*. The top gene features associated with CIMP were identified by feature selection function using L1-regularized logistic regression based on coefficient value. The expression profiles with all genes of the 288 (33 + 255) TCGA pRCC samples (one of the 34 samples does not have expression data) were used. The gene expressions were normalized across the samples using z-score for each gene feature. In the training process, L1-regularization was used for sparse coefficient values and the inverse regularization strength value was set as $C = 1.0$.

*Model for predicting CIMP*. Random Forest model was trained with expression of the origin-derived features and LDHA. The model was trained on the 33 classified pRCC samples (one of the 34 samples does not have expression data) using scikit-learn (V.0.4.2) package. The trained model was then used to predict CIMP from the other 255 pRCC samples. In the Random Forest model, we set the maximum depth as 2 to avoid overfitting.

*Gene set enrichment analysis*. Gene set enrichment analysis was performed on gene activity matrix of pRCC subgroups and their respective cell-of-origin cell types using GSEA[63] (4.1.0) against hallmark gene sets.

*Survival analysis*. Survival analysis of pRCC (TCGA) was performed using "Cgdsr" and "Survival" in R. The Kaplan-Meier method was used to generate curves for overall survival, which was defined as the time from the nephrectomy to death of any cause. The Log-rank test was used to assess the univariate survival differences.

For Hazard plots of risk of dying from pRCC, the Cox Proportional Hazard model included the binary value of feature gene expression, age, gender, stage, and tumor type as covariates. Feature gene expression was divided into "high" or "low" representing the upper or lower quarter of gene expression.

*Differential expression between CIMP and NotCIMP in pRCCa2_Like*. We filtered out genes with average expression < 1 or deviation of expression > 10 in expression matrix. The differential expression genes between CIMP and NotCIMP in pRCCa2_Like were identified as "abs (logFC) > 1, FDR < 0.05, logCPM > 5" using edgeR. Gene ontology enrichment was performed using Metascape[64] or KEGG pathways analysis. The rank matrix scores of genes in oxidative phosphorylation pathway were calculated by GSEA.

*Immunofluorescence staining*. Tissues from patients were formalin-fixed, paraffin-embedded, and processed for sectioning. For immunofluorescence staining, sections were deparaffinized, rehydrated and underwent antigen retrieval. Sections were then blocked with 10% bovine serum in PBS for 10 m and incubated with primary antibodies for 4 h at RT. After wash, sections were incubated with secondary antibodies for 1 h at RT and mounted with Fluoroshield Mounting Medium with DAPI (ab104139, Abcam). Images were obtained with fluorescence microscopy (DM5000B, Leica). The primary antibodies included mouse anti-AQP1 (ab9566; Abcam, 1:100 diluted), rabbit anti-GATA3 (ZA-0661; ZSGB-BIO), rabbit anti-GRHL2 (HPA004820; Sigma-Aldrich,1:50 diluted), rabbit anti-CK7 (ZA-0573; ZSGB-BIO) and mouse anti-CK7 (ZM-0071; ZSGB-BIO). The secondary antibodies included Alexa Fluor® 647 goat anti-rabbit IgG (H&L) (ab150079, Abcam), Alexa Fluor® 594 goat anti-rabbit IgG (B40944, Invitrogen), Swine Anti-Rabbit IgG (F020502-2, Dako), Rabbit Anti-Mouse IgG (F026102-2, Dako) and Alexa Fluor® 488 Donkey Anti-Mouse IgG (H&L) (ab150105, Abcam).

*Statistics*. Differential analysis of gene activities for feature genes between pRCC subtypes and their cell-of-origin was assessed using paired $t$ test (for pRCCa1 origin features: df = 24, $t = 8.9647$ between pRCC subtypes and $t = 18.927$ between origin cell types; for pRCCa2 origin features: df = 6, $t = −3.7814$ between pRCC subtypes and $t = −6.3796$ between origin cell types).

The probability of recovering all CIMP in pRCCa2_Like (41 samples) was tested by bootstrap. We randomly selected 41 samples in 255 pRCC samples and counted the number of CIMP. The processes were repeated 10000 times to generate a distribution of the number of recovered CIMP. We then calculated the probability of recovering all eight CIMP using "pnorm" with parameter "lower.tail = FALSE" based on this distribution. All other statistical analyses were described in respective sections.

**Reporting summary**. Further information on research design is available in the Nature Research Reporting Summary linked to this article.

## Data availability
The scRNA-seq publicly available data used in this study are available in the GEO database under accession code GSE140989[20], GSE121862[21], GSE131882[22] or downloaded from the supplementary files of Young et al.[19]. The publicly available ATAC-seq data for 34 pRCC samples and the cancer regulatory networks were downloaded from https://gdc.cancer.gov/about-data/publications/ATACseq-AWG. The publicly available gene expression (FPKM) of pRCC was downloaded from https://xenabrowser.net/datapages/?cohort=GDC%20TCGA%20Kidney%20Papillary%20Cell%20Carcinoma%20(KIRP)&removeHub=https%3A%2F%2Fxena.treehouse.gi.ucsc.edu%3A443. The publicly available clinical information of pRCC was downloaded from the supplementary files of Ricketts et al., 2018[2]. The raw scATAC-seq data generated in this study have been deposited in the GEO database under accession GSE166547. The raw scATAC-seq data are also available at Genome Sequence Archive for Human (GSA-Human) under accession number HRA001419. The remaining data supporting the findings of this study are available within the Article, Supplementary Information or Source Data file. Source data are provided with this paper.

## Code availability
Customized code[65] used in this study is available at GitHub (https://github.com/labYangNJU/scATAC-seq) and Zenodo and the corresponding DOI is as follows: (https://doi.org/10.5281/zenodo.5537024).

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

## Acknowledgements

We thank Dr. Si Wei for insightful discussion and thank Ai Lin Chun for critically reading and editing the manuscript. We are grateful to support from the National Natural Science Foundation of China (81500515, J.Y.), Natural Science Foundation of Jiangsu Province (BK20150591, J.Y.), Nanjing University and Emory University

Collaborative Research Grants (NE2019003, J.Y.), the Fundamental Research Funds for the Central University (021114380172, J.Y.), Jiangsu Clinical Medical Center (innovation platform, YXZXA2016003, Z.L.), The Open Project of Jiangsu Biobank of Clinical Resources (JSRB2021-01, J.Y.).

## Author contributions

J.Y. and Z.L. conceived of and designed the project. Q.W. and J.Y. wrote the manuscript with input from all other authors. Y.F., X.Z., K.Z., S.J., and H.G. prepared the para-cancerous and tumor samples. Y.Z. generated scATAC-seq libraries and performed the immunofluorescence staining experiments; Q.W. designed the data analysis plan and optimized Cicero. Q.W. and B.Z. performed data analysis with assistance from Z.Q., E.L., Y.X., and J.Z.. All authors read and approved the final manuscript.

## Competing interests

The authors have declared that no conflict of interest exists.

## Additional information

**Peer Review Information** *Nature Communications* thanks Ari Hakimi, Thomas Mitchell and the other, anonymous, reviewer(s) for their contribution to the peer review of this work. Peer reviewer reports are available.

