## [Peer Review File · Nature Communications]

Single-cell chromatin accessibility landscape in kidney identifies additional cell-of-origin in heterogenous papillary renal cell carcinomaReviewers' Comments:

Reviewer #1:

Remarks to the Author:

The authors should be applauded for their efforts to generate biological insights from newly generated single cell ATAC sequencing data. The main message, that the phenotypic heterogeneity of papillary renal cell carcinoma can be explained by different cells of origin is a significant deviation from our current understanding. As such, the theory requires a strong evidence to support this claim.

My main concern is that cellular reprogramming away from the cell of origin, be it from cellular dedifferentiation or metabolic divergence, is being misinterpreted as development from an entirely different cell of origin. Answering the following points may help alleviate these concerns.

1. In Figure 2b and c you compare a correlation coefficient of previously published chromatin accessibility in a number of tumours with your single cell data. Why are the single cell references split between the figures and no overall comparison used? Why not use a quantitative similarity score as per Young et al 2018? Without direct comparison between all possible reference cell types, the comparison as it stands is meaningless.

2. We know from previous reports that CIMP pRCC tumours are metabolically divergent, and are therefore more likely to transcriptionally less similar to their cell of origin. In all of the IHC examples, you have used GATA3 as an example of a CD-specific marker that is also upregulated in this subset of pRCCs. Unfortunately GATA3 is a well-established marker of de-differentiation and therefore perhaps not a good marker of similarity to differentiated adult cell state. Are there any specific markers of collecting function that are also upregulated in this subset of pRCC tumours?

With the above points addressed, there still needs to be some orthogonal evidence to support the unusual claims. It is far more likely that the more lethal variants of pRCC have deviated further transcriptionally from their original cell of origin.

Reviewer #2:

Remarks to the Author:

Wang et al perform scATACseq on 2 normal kidney tissue samples and reanalyze a data set of TCGA tumors that had atacseq performed to identify cell of origin of pRCC tumors. While the study is scientifically sound the findings of are of limited biologic and therapeutic significance.

Both the KIRP paper and the subsequent pan kidney TCGA projects (Chen et al Cell Reports 2016, Ricketts cell reports 2018) identified distinct metabolic, immunologic and signally features of the more aggressive "type 2 papillary tumors" thus significantly limiting the biologic relevance of these findings. Further many of the type 2 papillary tumors are known to be distal nephron tumors (by common IHC markers) especially the NF2 mutated tumor types. Would be useful to know the mutation status of these tumors predicted to common from distal tubules.

Chen et al in 2016 utilized RNAseq to identify cell of origin of all RCC TCGA tumors The authors should discuss and assess differences as the authors found the majority to be PCT in origin

Reviewer #3:

Remarks to the Author:

This study collects scATAC-seq data from normal human kidney tissues to infer the cell-of-origins in pRCC samples collected from TCGA, and reveals two populations of pRCC patients. The findings have potential importance for predicting clinical performance. This work is generally interesting and clearly presented. Below are several comments on the manuscript:

1. What's the overlap percentage in Figure 2a? The color bar does not clearly show what the actual number is.
2. Is it possible to combine Figure 2b and 2c to show a global correlation picture of all cell types? Correlation for PT groups can be highlighted on the global picture.
3. The authors suggested that 'pRCCa2 cell-of-origin might be a predictor for CIMP'. Is it possible to design a score based on pRCCa2 and evaluate its prediction performance of CIMP (e.g. AUROC)? Likewise for the clinical association analysis in Figure 4?
4. Is there any justification why the significance of overlap is calculated in the way described as line 404? Will a permutation test more appropriate in this case?
5. It seems that non of the p-values reported in the paper are adjusted for multiple testing. Plus, the cutoff of statistical significance seems to be quite arbitrary. For example, the cutoff is 0.001 on line 404 and becomes 0.01 on line 429,451 and 488. Consider redo the statistical significance with a more rigorous manner.
6. The whole paper is based on the clustering of normal human kidney cells. Is the clustering result robust and consistent if other scATAC-seq clustering algorithms are used, such as scABC, cisTopic, SCRAT, chromVAR?

REVIEWER COMMENTS

Reviewer #1, expert in kidney cancer biology and developmental origin and sc-RNAseq (Remarks to the Author):

The authors should be applauded for their efforts to generate biological insights from newly generated single cell ATAC sequencing data. The main message, that the phenotypic heterogeneity of papillary renal cell carcinoma can be explained by different cells of origin is a significant deviation from our current understanding. As such, the theory requires a strong evidence to support this claim.

My main concern is that cellular reprogramming away from the cell of origin, be it from cellular dedifferentiation or metabolic divergence, is being misinterpreted as development from an entirely different cell of origin. Answering the following points may help alleviate these concerns.

1. In Figure 2b and c you compare a correlation co-efficient of previously published chromatin accessibility in a number of tumours with your single cell data. Why are the single cell references split between the figures and no overall comparison used? Why not use a quantitative similarity score as per Young et al 2018? Without direct comparison between all possible reference cell types, the comparison as it stands is meaningless.

We thank the reviewer for this comment. In the literature, it is believed that proximal tubular (PT) cells are the origin of papillary renal cell carcinoma (pRCC). Therefore, in the original manuscript, we first split the reference and focused on the correlation of chromatin accessibility between pRCC samples and the PT cells. Following the reviewer's advice, we now remove the comparison on split reference, and provide the overall comparison instead.

In the revised manuscript, we provided overall comparison between pRCC samples and all reference cell types using two different methods: the correlation coefficient and the similarity score. Correlation coefficients between pRCC samples and each of the reference cell types are presented in the revised **Figure 2b**.

We calculate similarity scores following Young et al. 2018. To calculate similarity scores between bulk pRCC ATAC-seq data and normal kidney scATAC-seq data, we generated pseudo-bulk reference panel from kidney scATAC-seq data. We randomly sample 50 cells 20 times from each cell type to generate a pseudo-bulk reference panel. We then trained logistic regression models on the cellular identities using chromatin accessibility features. These models were used to calculate similarity scores for each pRCC sample. The details about calculating similarity scores are described in the 'Comparison by similarity score' section in the Methods part. The similarity scores confirm that there are two distinct groups among all pRCC samples (**Figure 2c**). A group of four pRCC samples found by the similarity score method are exactly the same four samples found to be distinct from PT cells based on correlation. The similarity scores clearly reveal that these four pRCC samples resemble CD_PC, which cannot be well detected by overall correlation coefficient profile. We want to thank the reviewer for the great suggestion.

2. We know from previous reports that CIMP pRCC tumours are metabolically divergent, and are therefore more likely to transcriptionally less similar to their cell of origin. In all of the IHC examples, you have used GATA3 as an example of a CD-specific marker that is also upregulated in this subset of pRCCs. Unfortunately GATA3 is a well-established marker of de-differentiation and therefore perhaps not a good marker of similarity to differentiated adult cell state. Are there any specific markers of collecting function that are also upregulated in this subset of pRCC tumours?

In addition to GATA3, in the revised manuscript, we also examined GRHL2 which is required for the function of collecting duct¹. We found that the chromatin at *GRHL2* is accessible in CD_PC cells and pRCCa2 samples, but not in PT cells or pRCCa1 samples (**Figure 2e**). Similar to GATA3, staining of GRHL2 showed that GRHL2 is not expressed in most of the pRCC samples (**Figure 2f**). However, there is a small group of pRCC samples that are GRHL2 positive and AQP1 negative (**Figure 2g**). These results confirm that pRCCs are heterogeneous in terms of marker genes of normal kidney cell types.

3. With the above points addressed, there still needs to be some orthogonal evidence to support the unusual claims. It is far more likely that the more lethal variants of pRCC have deviated further transcriptionally from their original cell of origin.

We thank the reviewer for the insightful comment. We here show evidences that the more lethal pRCC variants are not only transcriptionally deviated from PT, but they also resemble CD_PC in multiple ways. We do not think these variants of pRCC display multitude CD_PC features by chance. Instead, we believe these variants of pRCC are originated from a cell-of-origin different from PT.

First, we investigated the transcriptional profiles of CIMP, the more lethal variants of pRCC. We identified the top gene features of CIMP using logistic regression model (details are described in 'Top gene features associated with CIMP' section in Methods), then we examined the expression of these top gene features in normal kidney cell types. The result showed that the top gene features associated with CIMP are lowly expressed in PT cells. We agree that this finding could be explained by transcriptional deviation from the PT origin. However, upon further investigation, we found these top gene features are highly expressed in CD_PC cells (**Figure S13**), a fact that is unlikely to occur due to transcriptional deviation from PT origin alone.

Second, we compared the chromatin accessibility pattern between the less lethal pRCCa1 group and the more lethal pRCCa2 group. Among all the kidney cell types, we found that the pRCCa1-specific chromatin-accessible regions show the highest chromatin accessibility in PT cells, and the pRCCa2-specific chromatin-accessible regions show the highest chromatin accessibility in CD_PC cells (**Figure 2d**). Additionally, the top TF-binding motif enriched in pRCCa1-specific chromatin-accessible regions is HNF4A, while the top TF-binding motif enriched in pRCCa2-specific chromatin-accessible regions is JUN-AP1 (**Figure S7**). It is notable that HNF4A is the top TF-binding motif enriched in PT-specific chromatin-accessible regions, and JUN-AP1 is the top TF-binding motif enriched in CD_PC-specific chromatin-accessible regions (**Figure 1h**).

Furthermore, some of the pRCC samples that we classified as pRCCa2_Like with CD_PC origin stay at low stage currently. These low stage pRCC samples also highly express GATA3 and GRHL2 and present CD_PC features. This finding cannot be explained by the hypothesis about pRCC lethality and transcriptional deviation from the PT origin.

Given all the evidences shown above, together with the similarity score and staining of cell type markers, we do not think the more lethal pRCC variants are those simply evolved further away from PT origin, and display all the CD_PC features by chance. We believe the heterogenous pRCCs are more likely to originate from different cell-of-origins.

Reviewer #2, expert in pRCC subtypes (Remarks to the Author):

Wang et al perform scATACseq on 2 normal kidney tissue samples and reanalyze a data set of TCGA tumors that had atacseq performed to identify cell of origin of pRCC tumors. While the study is scientifically sounds the findings of are of limited biologic and therapeutic significance.

1. Both the KIRP paper and the subsequent pan kidney TCGA projects (Chen et al Cell Reports 2016, Ricketts cell reports 2018) identified distinct metabolic, immunologic and signally features of the more aggressive "type 2 papillary tumors" thus significantly limiting the biologic relevance of these findings.

We thank the reviewer for the comment. As the reviewer correctly pointed out, previous studies^{2, 3} have identified features like metabolic or immunologic features for CIMP--the more lethal pRCCs. However, it is unclear whether these features are the cause or consequence of tumour progression. For example, ccRCC aggressiveness is linked to increases in metabolites involved in glutathione and cysteine/methionine metabolism pathways⁴. These features would be dynamic during tumour progression. Although they indicate the more lethal state of pRCC, they are probably unable to serve as prognostic biomarkers for early detection of the potential aggressive pRCC.

In contrast, cell-of-origin is stable and retained in the evolving tumor cells, therefore, it is a much more reliable prognostic biomarker for early diagnosis. Our results suggest a type of metabolic rewiring could lead to aggressive state only in the CD_PC-originated pRCC (**Figure 4i**). Thus, cell-of-origin is an independent and important indicator of pRCC subtypes. Integration of cell-of-origin and other biomarkers such as metabolic features could facilitate early prediction and monitoring of the aggressive pRCC. Therefore, in our opinion, our findings are of significant biologic and therapeutic significance. In the revised manuscript, we added discussion of this important point in both introduction and discussion.

2. Further many of the type 2 papillary tumors are known to be distal nephron tumors (by common IHC markers) especially the NF2 mutated tumor types.

Would be useful to know the mutation status of these tumors predicted to common from distal tubules.

We thank the reviewer for the insightful suggestion. In the revised manuscript, we examined the somatic mutation status of pRCCs using TCGA data. For the 11 significantly mutated genes (SMGs) reported for pRCC⁵, we found mutations in MET, PBRM1 and SMARCB1 only exist in pRCC with PT origin (pRCCa1_Like) (**Figure S11a**). This is consistent with literature report that MET alterations are associated with Type 1 pRCCs⁶, and consistent with our finding that most Type 1 pRCCs are classified as pRCCa1_Like.

As the reviewer pointed out, mutations in NF2 are enriched in pRCCs originated from distal tubules (pRCCa2_Like) (**Figure S11b**). Among the 11 reported SMGs, only NF2 is found to be enriched in pRCCa2_Like ($p=0.0486$). We then look at all the detected mutations and identify genes with mutations enriched in pRCCa2_Like (**Figure S11c**). Among the 41 genes identified, mutations in FH have previously been associated with CIMP⁶. We further found these 41 genes are enriched in NFR2 and HIPPO pathways (p -values are 0.001 and 0.0007 respectively) (**Figure S11d**). Previously, Chen et al. 2016 found transcriptional targets of NFR2 and HIPPO pathways were elevated in CIMP, but except for NF2, no gene mutation can be found to account for the observed transcriptional differences among subtypes². With our cell-of-origin classification, we found mutations in TXNRD1, SLC6A14 and SLC5A7 in the NFR2 pathway and ITGAL, KRAS and NF2 in the HIPPO pathway are enriched in pRCCa2_Like subgroup which contains CIMP (**Figure S11d**). However, Given the scarcity of mutations in these genes, without additional information, no mutation or mutation combination can predict the pRCC subgroup with different cell-of-origins (**Figure S11e**).

3. CHen et al in 2016 utilized RNAseq to identify cell of origin of all RCC TCGA tumors The authors should discuss and assess differences as the authors found the majority to be PCT in origin

Thanks for the reviewer's comment. Chen et al. 2016 examined the cell-of-origin of all RCC by comparing the transcriptome of RCC to the transcriptome of normal kidney at a resolution of microdissected nephron, and found pRCC cases resemble the proximal nephron². The result from Chen et al. 2016 showed that some of the pRCC samples do not look similar to the proximal nephron, which is not mentioned in their paper. However, because the resolution of the transcriptome in Chen et al. is not high enough, it is impossible to resolve the cell-of-origin of those pRCCs at the cell type level. As genetic lesions or transformation resulting from renal insult for tumor initiation is cell type-specific and context-dependent, it is necessary to identify the cell-of-origin of renal cell carcinoma at the cell type resolution. In this study, single cell ATAC-seq profiles enable us to investigate the potential heterogeneous cell-of-origin of pRCCs at cell type level. In addition to confirming that the majority of pRCCs are PT-origin, we discovered that pRCCs can also originate from the collecting duct principal cells of kidney. Furthermore, we showed that pRCCs with different cell-of-origins exhibit different molecular characteristics, cell transformation and clinical behaviours. In the revised manuscript, we added discussion about these points.

Reviewer #3, expert in sc-ATACseq (Remarks to the Author):

This study collects scATAC-seq data from normal human kidney tissues to infer the cell-of-origins in pRCC samples collected from TCGA, and reveals two populations of pRCC patients. The findings have potential importance for predicting clinical performance. This work is generally interesting and clearly presented. Below are several comments on the manuscript:

1. What's the overlap percentage in Figure 2a? The color bar does not clearly show what the actual number is.

We have revised the figure to present the actual number in the color bar (**Figure 2a**). The number of cell type-specific peaks range from 439 to 3,774, and the number of cancer type-specific peaks range from 11,819 to 49,748. The actual percentage of cancer type-specific peaks that overlap with cell type-specific peak is 0.0058% to 7.4%. We also look at the percentage of cell type-specific peaks that overlap with cancer type-specific peaks (**Figure S4b**). The actual percentage is 0.58% to 31.3%, and the conclusion is the same as PT cell type-specific peaks are enriched in renal cell carcinoma.

2. Is it possible to combine Figure 2b and 2c to show a global correlation picture of all cell types? Correlation for PT groups can be highlighted on the global picture.

Thanks for the reviewer's suggestion. We now present the global correlation picture of all cell types together, and highlight the PT groups in **Figure 2b**.

3. The authors suggested that 'pRCCa2 cell-of-origin might be a predictor for CIMP'. Is it possible to design a score based on pRCCa2 and evaluate its prediction performance of CIMP (e.g. AUROC)? Likewise for the clinical association analysis in Figure 4?

We trained a model on pRCCa2 samples with our discovered origin-derived features together with the metabolic reprogramming feature LDHA using Random Forest classifier (details are described in 'Model for predicting CIMP' section in Methods). Using this model, we are able to identify CIMP from the 255 pRCC samples in TCGA. The AUROC is 0.98 (**Figure S16**). We thank the reviewer for the great suggestion. This model improves our prediction on pRCCs.

4. Is there any justification why the significance of overlap is calculated in the way described as line 404? Will a permutation test more appropriate in this case?

We thank the reviewer for the suggestion. Following the reviewer's advice, we performed permutation test to evaluate the significance of the overlaps. For each overlap between a cell type and a cancer type, we randomly select the same number of peaks for the cell type from the pooled cell type-specific peaks, count the number of the cancer type-specific peaks that overlap with the randomly select cell type-specific peaks, and calculate the percentage. We repeat these steps 1000 times, and the average of the 1,000 percentages is taken as the expected overlapping

percentage. p-value is calculated using binomial test with the `binom.test` function in R. All the p-values are then adjusted using the Benjamini–Hochberg procedure with the `p.adjust` function in R. The results show that renal cell carcinoma including pRCC and ccRCC show significant overlap with kidney PT cells (FDR as $1e-216$, $1e-300$ respectively) (**Figure S4a**).

5. It seems that non of the p-values reported in the paper are adjusted for multiple testing. Plus, the cutoff of statistical significance seems to be quite arbitrary. For example, the cutoff is 0.001 on line 404 and becomes 0.01 on line 429,451 and 488. Consider redo the statistical significancy with a more rigorous manner.

We thank the reviewer for the suggestion. In the revised manuscript, we have adjusted multiple testing in the significance tests. For the significance of overlapping between kidney cell type-specific peaks and cancer type-specific peaks described on line 404 of the original manuscript, we now calculated p-values from permutation test and then corrected all the p-values to false discovery rate using the Benjamini–Hochberg method in the revised manuscript. For the differential peaks between pRCC subgroups on original line 429, feature genes between pRCC subgroups on original line 451, and differential expressed genes between CIMP and Not-CIMP in pRCCa2_Like on original line 488, multiple testing was corrected using the Benjamini–Hochberg method, and $FDR < 0.05$ was used instead of $p\text{-value} < 0.01$ in the revised manuscript.

Using the new significance threshold, the significance result of peak overlap is shown in **Figure S4**, and the overlap between renal cancer type-specific peaks and PT-specific peaks is still highly significant. The differential peaks between pRCC subgroups are shown in **Figure S7**, and the motif enrichments of the identified differential peaks remain the same. Although fewer feature genes are identified in each pRCC subgroup, the genes overlapping with origin-feature-genes remain the same as listed in **Figure S10a**. Despite the fact that fewer genes are identified as significant differentially expressed genes between CIMP and Not-CIMP, the oxidative phosphorylation pathway is still the top enriched term on the new gene list as showed in **Figure S16a**. All the findings remain the same when using the threshold after correction for multiple testing.

6. The whole paper is based on the clustering of normal human kidney cells. Is the clustering result robust and consistent if other scATAC-seq clustering algorithms are used, such as scABC, cisTopic, SCRAT, chromVAR?

We thank the reviewer for the comment. In the revised manuscript, we have performed additional analysis using a different scATAC-seq clustering algorithm named cisTopic. In a recent survey of single-cell ATAC-seq data analysis methods⁹, cisTopic is found to be the second-best method following SnapATAC. Therefore, we choose cisTopic algorithm to repeat the analysis.

Using cisTopic, we can still identify the main kidney cell types and the clustering result is robust (**Figure S5**). Based on the clustering result of cisTopic, we examine the correlation coefficients and similarity scores of pRCC samples and normal kidney cell types. There are four pRCC samples that are distinct from PT cells (**Figure S6a**) and all four samples show high similarity to CD-PC (**Figure S6b**). These are the same four samples discovered in the analysis conducted using SnapATAC

clustering. All the conclusions in the original analysis remain unchanged when conducted clustering using cisTopic instead.

1. Hinze C, *et al.* GRHL2 Is Required for Collecting Duct Epithelial Barrier Function and Renal Osmoregulation. *J Am Soc Nephrol* **29**, 857-868 (2018).
2. Chen F, *et al.* Multilevel Genomics-Based Taxonomy of Renal Cell Carcinoma. *Cell Rep* **14**, 2476-2489 (2016).
3. Ricketts CJ, *et al.* The Cancer Genome Atlas Comprehensive Molecular Characterization of Renal Cell Carcinoma (vol 23, pg 313, 2018). *Cell Reports* **23**, 3698-3698 (2018).
4. Hakimi AA, *et al.* An Integrated Metabolic Atlas of Clear Cell Renal Cell Carcinoma. *Cancer Cell* **29**, 104-116 (2016).
5. Lawrence MS, *et al.* Discovery and saturation analysis of cancer genes across 21 tumour types. *Nature* **505**, 495-+ (2014).
6. Linehan WM, *et al.* Comprehensive Molecular Characterization of Papillary Renal-Cell Carcinoma. *New England Journal of Medicine* **374**, 135-145 (2016).
7. Young MD, *et al.* Single-cell transcriptomes from human kidneys reveal the cellular identity of renal tumors. *Science* **361**, 594-+ (2018).
8. Corces MR, *et al.* The chromatin accessibility landscape of primary human cancers. *Science* **362**, 420-+ (2018).
9. Chen H, *et al.* Assessment of computational methods for the analysis of single-cell ATAC-seq data. *Genome Biol* **20**, 241 (2019).

Reviewers' Comments:

Reviewer #1:

Remarks to the Author:

I am satisfied that the revisions made by the authors have addressed my original comments.

Reviewer #2:

Remarks to the Author:

satisfied with responses to critiques.

Reviewer #3:

Remarks to the Author:

The authors have addressed all my comments.

REVIEWERS' COMMENTS

Reviewer #1 (Remarks to the Author):

I am satisfied that the revisions made by the authors have addressed my original comments.

Reviewer #2 (Remarks to the Author):

satisfied with responses to critiques.

Reviewer #3 (Remarks to the Author):

The authors have addressed all my comments.

We thank all the reviewers for the insightful comments and suggestions. We are glad that our revisions resolve all the questions properly.